
# Iron oxides control sorption and mobilisation of iodine in a tropical rainforest catchment

Laura Balzer[1]*, Katrin Schulz[2], Christian Birkel[3], Harald Biester[1]

[1]Department of Environmental Geochemistry, Technical University, Braunschweig, 38106, Germany
[2]Soil Chemistry Group, Institute of Biogeochemistry and Pollutant Dynamics, CHN, ETH, Zürich, 8092, Switzerland
[3]Department of Geography and Water and Global Change Observatory, University of Costa Rica, San José, Costa Rica

*Correspondence to*: Laura Balzer (laura.balzer@tu-braunschweig.de)

**Abstract.** Iodine is an essential trace element for all mammals and its bioavailability in terrestrial systems depends on its accumulation in soils but also on its release into the aquatic system. In tropical systems retention and mobilization of iodine in soils and related concentrations in streamflow are poorly understood. We, therefore, investigated the relationship between solid phase iodine binding on hillslope soils and the iodine and dissolved organic carbon (DOC) mobilisation to a connected stream. Our study was conducted in a pristine pre-montane rainforest in Costa Rica with old (up to nine My) and highly weathered volcanic soils. A total of nine soil profiles from two tributary sub-catchments to the main stream were sampled. Solid phase sequential extraction was used to identify iodine binding forms in soils. The water leachable iodine fraction was additionally assessed by batch soil leaching experiments. Stream water was sampled randomly over a period of five weeks. Results showed extremely high iodine concentrations in soils, ranging from 53-130 mg kg$^{-1}$ (median: 69 mg kg$^{-1}$), which is 13-fold higher than in temperate soils. In contrast, median water-soluble iodine was only 0.01 % (range: 0-0.4 %) of total soil iodine. Solid phase sequential extractions revealed iodine sorption to iron oxides (median: 79 % of total iodine) as the main retention factor. High enrichment and low mobilisation of iodine in soils caused relatively low iodine concentrations (0.77-1.26 µg L$^{-1}$) in stream waters during base and even moderate high flow conditions. The significant correlation of iodine and DOC in soil leachates suggested transport of organically bound iodine from upper- to subsoil horizons and strong sorption of DOC-iodine complexes to iron oxides. Our results showed that the old and highly weathered tropical soils in the study area were highly enriched in iodine caused by strong retention of DOC bound iodine to iron oxides. As a result, iodine release from soils was low which resulted in low stream water iodine concentrations and subsequently in a low bioavailability.

## 1 Introduction

Iodine is a halogen essential for mammals due to its function in the synthesis of thyroid hormones (Andersson et al., 2007). A lack of iodine leads to several health problems, referred to as iodine deficiency disorders (IDDs), such as goitre and cretinism (Andersson et al., 2007). IDDs are common in mountain regions and is of particular concern in tropical areas (Dissanayake et al., 1999).



The majority (>70 %) of iodine is stored in the oceans (Hou et al., 2009). Oceanic iodine compounds are released to and transported in the atmosphere and deposited by wet and dry depositions to terrestrial and aquatic ecosystems (Shetaya et al., 2012). The iodine content in the parent rock is generally significantly lower than in the overlying soil (Whitehead, 1984). This means that the atmospheric deposition derived from the ocean is the main iodine input to soils. Concentrations of iodine in

soils vary largely depending on soil type and distance from the ocean (Whitehead, 1984; Fuge and Johnson, 1986). According to Whitehead (1984) typical iodine concentrations in soils worldwide vary between 0.5 and 20 mg kg$^{-1}$. Soils located near the coast show higher iodine concentrations than soils further inland (Fuge and Johnson, 1986). Concentrations up to 660 mg kg$^{-1}$ were found in an organic-rich soil near the coast (Smyth and Johnson, 2011). Soils rich in organic matter (OM) (Xu et al., 2011a; Xu et al., 2011b), clay minerals and/or sesquioxides (Yoshida et al., 1992; Shetaya et al., 2012) have a high iodine

fixation potential (IFP). Particularly the formation of organically bound iodine (Org-I) and sorption of Org-I to soil particles plays an important role in iodine fixation (Hou et al., 2009; Hou et al., 2003; Muramatsu et al., 2004; Yoshida et al., 1992; Yeager et al., 2017; Dai et al., 2009; Shetaya et al., 2012; Fuge and Johnson, 1986; Schwehr et al., 2009; Whitehead, 1984, 1973). Due to the affinity of iodine to bind to OM, the mobility of iodine in terrestrial environments is closely related to that of dissolved organic carbon (DOC) (Xu et al., 2011a; Gilfedder et al., 2010). Considerable amounts of iodine can also be

mobilised as colloidal Org-I (Xu et al., 2011b). However, iodine mobility and in turn bioavailability is largely dependent on the soil IFP.

Even though numerous previous studies investigated iodine soil chemistry, there are only few conducted in tropical ecosystems and to the best of our knowledge there is no field study focusing on iodine soil chemistry including iodine binding forms in

the tropics. The tropics are characterized by high temperatures with little intra-annual variability (minimum air temperature ≥18 °C (Kottek et al., 2006)) and high rainfall (minimum monthly precipitation ≥60 mm (Kottek et al., 2006)), resulting in intense chemical weathering (e.g. Porder et al. (2007); Cheng et al. (2017)). Tropical soils are highly weathered acidic soils with high accumulation of iron-oxides (Fe-oxides) (e.g. Aristizábal et al. (2005)) due to their higher age, compared to temperate soils, since tropical soils developed mostly without the influence of recent glaciations. Since DOC has a strong affinity to sorb

to sesquioxides (Lilienfein et al., 2004) and iodine has a high affinity to sorb to DOC, high Fe- oxide content may increase the IFP and decrease the iodine release from soils to the aquatic systems. We hypothesise that tropical soils high in OM and Fe-oxides in regions with high annual precipitation constitute a potential sink for atmospheric derived iodine.

Therefore, this work deciphers soil-related factors that dominate iodine retention and mobilisation in tropical soils formed

more than five million years ago with the following objectives: 1) Determine the iodine mobility in tropical soils by means of water-soluble iodine in batch leaching experiments and comparison of iodine concentration in the draining stream at different flow conditions and 2) Identify the main iodine binding forms in soils by sequential extraction (SE).



## 2 Materials and Methods

### 2.1 Study site

The study area is located in the Alberto Manuel Brenes Biological Reserve (ReBAMB) managed by the University of Costa Rica, which forms part of the Central Volcanic Cordillera in Costa Rica, Central America. The San Lorencito river catchment
has an area of around 3.2 km², an elevation gradient from 890 to 1450 masl (Figure 1) and drains towards the Atlantic coast (Caribbean Sea). The study area has been formed more than five million years ago in the Tertiary-Neogene period by volcanic activity, soil formation and erosion. The geology consists of basaltic and andesitic rocks overlain by acidic and porous clayey-silty Cambisol soils (REF to the classification system).

The catchment is characterized by steep, V-form valleys with a mean slope of 17°. The main channel has a length of almost
3.2 km and a mean river slope of 6.3°. The vegetation represents typical pristine pre-montane tropical rainforest with a maximum height of 50 m. The mean annual precipitation is around 2800 mm yr$^{-1}$ (2010 to 2018), the potential evapotranspiration is relatively low around 500 mm yr$^{-1}$. The mean annual air temperature is close to 21 °C with a constantly high relative humidity of 98 %. The dry season with lower rainfall probability lasts from January until April followed by the rainy season between May and December. See Dehaspe et al. (2018) and Solano-Rivera et al. (2019) for a more complete
description of the study catchment.





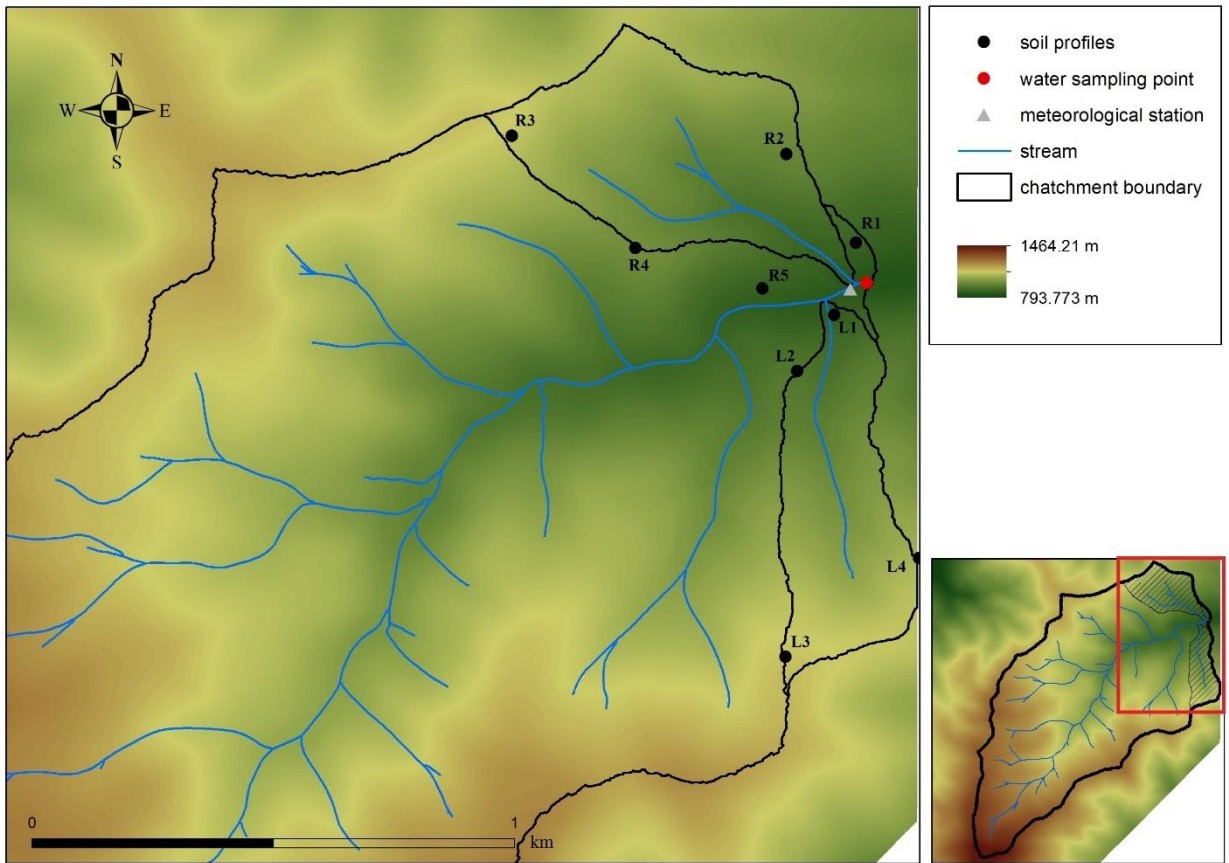

**Figure 1: Topography of the San Lorencito catchment, ReBAMB, Costa Rica. Left: Detailed map of the study area showing the locations of sampled soil profiles (black points); the sampling point for stream water (red point). The boundaries of the catchments are shown in black.**

## 2.2 Field work and sampling

Soil samples were taken from nine soil profiles on four consecutive days in June 2017, at the beginning of the rainy season. The sites for the profiles were evenly distributed around two tributaries of the main channel (Figure 1). Looking upstream, five soil profiles were established on the right hillslope (R1-R5) and four on the left hillslope of the main stream (L1-L4). The profile depth was 0.5 m-0.6 m. One soil profile with a depth of 1 m was established in the middle of each slope (R2, L2). Profiles L1, R1, R4 and R5 showed a clear distinction between topsoil (Ah), found in the upper 5 cm of the soil profile and subsoil (Bw) horizon (≥20 cm), whereas profiles L3, L4, R2 and R3 showed a transitional (AhBw) horizon in-between (15-20 cm). In profile L2 we found a second subsoil layer (II Bw) that might have been buried by eroded soil material. At each soil profile, 0.5 kg of disturbed soil material was taken from the middle of each horizon, stored in PE bags (Whirl Packs) and shipped to Germany for further analyses. Parent rock (andesite and basalt) samples were taken from accessible intact rocks in the catchment.





Physical soil properties for each horizon were recorded in the field including texture, aggregate type, degree of rooting, skeleton and pore volume (see Supporting Information).

In May and June 2017, 14 water samples from the main stream were taken every one or two days in four intervals. Samples were taken using pre-rinsed 50 ml PE Falcon tubes. The frequency and duration of each sampling interval was dependent on
accessibility of the study site and intensity of rain events. The total period of sample collection was five weeks.

## 2.3 Analytical methods

The pH value of the air-dried soil was measured in water with a soil:solution ratio of 1:2.5. The water content ($\theta$) was determined gravimetrically for all horizons of profiles L2 and R2. Stream water temperature, pH, redox potential (Eh) and conductivity (EC) were measured in-situ using a handheld Hanna multi-parameter probe (HI 98195). Discharge was
determined using a digital water velocity meter (FP111 flow probe; Global Water). High resolution water level values were obtained from a multi-sensor box located at the main channel (Water Quality Monitoring System; Global Water). Precipitation data was measured as canopy throughfall from a meteorological station inside the rainforest close to the stream at 1171 masl (climate monitoring station; Global Water) and was sampled by means of a precipitation gauge (Rain sampler RS-1B, Palmex-Zagreb) during one week (07.-14.06.2017). Both streamflow and meteorological stations recorded data at five-minute intervals
during the entire sampling period. All water samples were vacuum filtered (<0.45 µm) using a nylon filter and stored frozen (-18 °C) prior to analysis.

### 2.3.1 Solid phase sequential extraction of iodine

To differentiate between various solid phase binding forms of iodine in the soil, a modified SE procedure according to Schmitz and Aumann (1995) was applied to the samples of the profiles L1, L3, R1, R3 and R5. After the soil samples were dried at
40 °C for three days, sieved to <2 mm and homogenized, a subsample of 25 g was used for the SE. The method consisted of four steps to distinguish water-soluble iodine, exchangeable iodine, iodine adsorbed on Fe-oxide surfaces and Org-I (Figure 2).





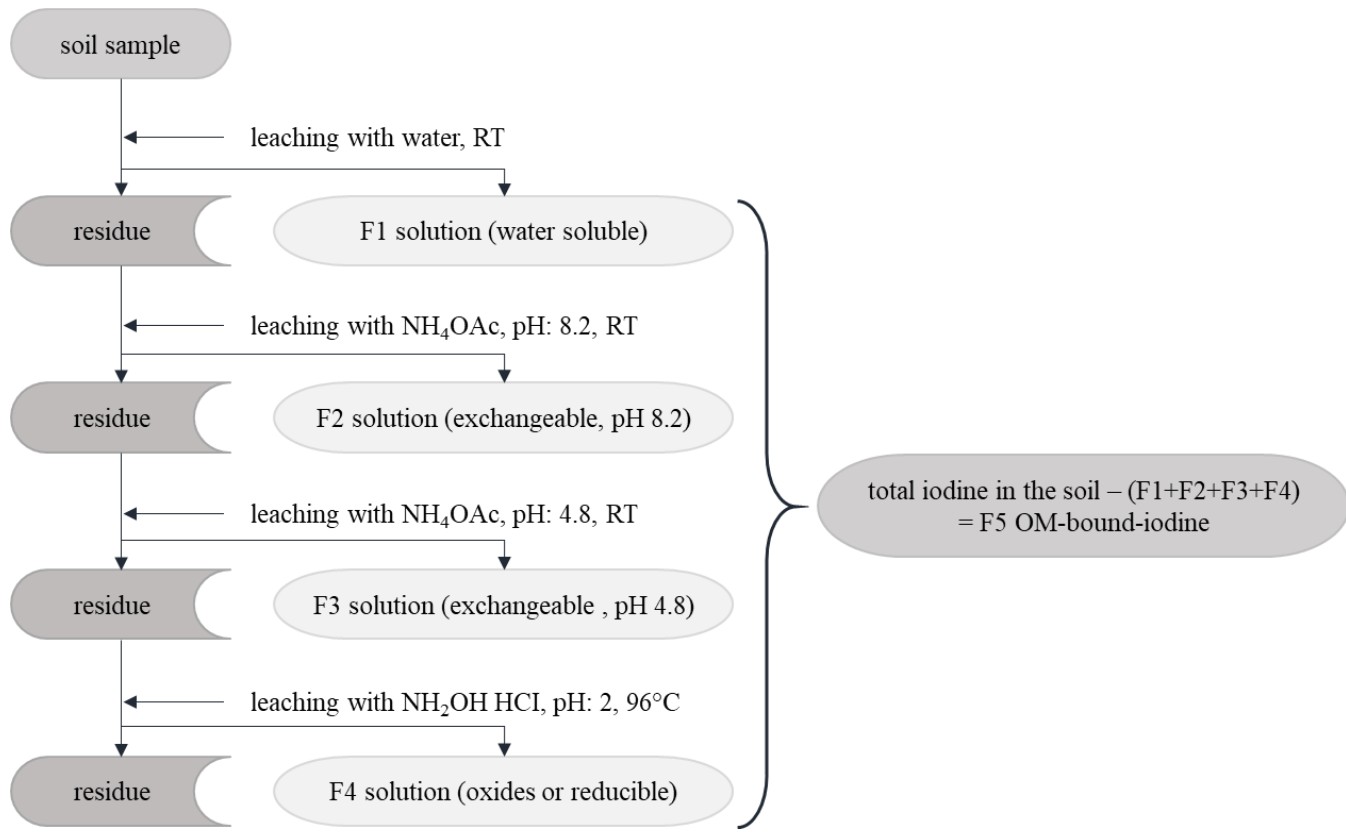

**Figure 2: Schematic diagram of the sequential extraction procedure for iodine fractionation in soil samples (after Hou 2009), with iodine fractions F1 to F5 (RT: room temperature).**

The first step (water) was the extraction of water-soluble iodine (F1). The second and third extraction steps extracted the exchangeable iodine using 1 M ammonium acetate ($NH_4OAc$) solution at pH 8.2 (F2) and at pH 4.8 (F3). In the fourth extraction step (F4), the soil samples were leached with 0.04 M hydroxylammonium-chloride ($NH_2OH\,HCl$) dissolved in 25 % (v/v) AcOH solution (F4) at pH 2 to release iodine adsorbed to Fe-oxides through the dissolution of the Fe-oxides. Step F5

10 targeting iodine bound to OM was determined indirectly by subtraction of the sum of the amount of iodine extracted during F1-F4 from the total amount of iodine in the solid soil samples. In all extraction steps a soil:solution ratio of 1:10 was used. The suspension was shaken moderately for 24 hours prior to centrifugation and decantation. Extraction F1-F3 were shaken at room temperature, extraction F4 at 96 °C. All suspensions were vacuum filtered (<0.45 µm) using nylon filters and stored frozen (-18 °C) until analyses.



### 2.3.2 Batch water leaching experiments

The total water leachable fractions of iodine, bromine (Br) and DOC in the soil samples were determined by batch leaching experiments with fresh soil samples. An aliquot of 10 g of fresh homogenized soil material was mixed with 100 ml of water, shaken for 24 hours and filtered using nylon filters (<0.45 µm). Compared to the solid phase SE, the batch leaching with fresh

soil material provides a more realistic assessment of the effect of DOC on iodine leachability from the soils.

### 2.3.3 Liquid phase analysis of iodine, Br, DOC and Fe

Liquid water samples from the stream, canopy throughfall, leachates of the SE- and batch leaching experiment were defrosted overnight at room temperature and analysed for iodine and Br by means of ICP-MS (Agilent 7700). The DOC concentration was determined by thermo-catalytic oxidation by means of a TOC-Analyser (multi N/C 2100, Analytic Jena) after acidification

of samples with 1 v/v HCl (2 %) to a pH of 2 to remove carbonic acid. Measurements of DOC were validated by comparison with certified reference samples (Mauri 09 and TOC20), both references with a mean recovery of 106 %.

The dissolved Fe concentration was determined using an ICP-MS (Agilent 7700; Germany) in acidified stream water and batch leaching samples (1 v/v $HNO_3$; 60 %). The quality of the measurements was controlled by a river water reference material (LGC6019) with a Fe concentration of $287\pm7$ µg L$^{-1}$. Deviation of measured Fe concentrations from the reference was < 1.6 %

and always within the given standard deviation.

### 2.3.4 Solid phase analysis

Soil and parent rock samples were freeze dried (LYOVAC GT 2-E) until constant weight and ground in an agate ball mill. Total iodine and Br analyses were carried out by thermal extraction using an AOX analyser (Thermo Euroglas AOX), with subsequent trapping in water (Gilfedder et al., 2008) followed by ICP-MS analyses. The detection limit for iodine and Br was

0.3 µg L$^{-1}$ and 0.4 µg L$^{-1}$ respectively. The quality of the measurements was controlled by a sediment reference material (DC 73312) with a certified iodine and Br concentration of $2.9 \pm 0.4$ mg kg$^{-1}$ and $3.0\pm 0.6$ mg kg$^{-1}$, respectively. Mean relative standard deviations (RSDs) were 6.6 % and 17.9 %, respectively (n=3). The carbon (C) content in soils was measured using an elemental analyser (EuroEA 3000). The quality of the measurements was controlled by a sediment reference material (DC 73312) with a C concentration of 4.2 g kg$^{-1}$. The recovery was 101.9 %.

Fe concentrations in soil were determined using an energy-dispersive x-ray fluorescence spectrometer following the method of Cheburkin and Shotyk (1996) using triplicates. The quality of the measurements was controlled by the reference material LKSD4 with a certified Fe concentration of $28.68 \pm 2$ g kg$^{-1}$ (RSD= 4.7 %, n=4).





## 2.4 Statistical analysis

The Spearman correlation coefficient ($r_S$) was used to identify relationships between iodine, Br and C (in solid samples) and DOC (in leachate samples). All statistical analyses and plots were performed using the statistical program R 3.2.1 (R Development Core Team 2015) except for the average, median, standard deviation, minimum and maximum values calculated using Excel (Microsoft Office, 2013). Schematic diagrams were visualized using Visio (Microsoft Visio Professional, 2013).

## 3 Results and Discussion

### 3.1 Total iodine in soils, parent rock and throughfall

Iodine concentrations in the soil samples ranged between 53 and 130 mg kg$^{-1}$ (median: 69 mg kg$^{-1}$), which is 13 times higher than mean iodine concentrations in soils of temperate latitudes (~5 mg kg$^{-1}$; Johnson, 2003). In Japanese Andosols, high iodine concentrations of up to 150 mg kg$^{-1}$ were attributed to ocean proximity, high rainfall (2000 mm yr$^{-1}$) and high adsorption capacities of soils (Muramatsu et al. 2004). Thus, iodine concentrations in the soils of the study area were in the upper range of observed concentrations elsewhere. Iodine enrichment during weathering could be neglected, because the iodine content in the parent rock was below 1.2 mg kg$^{-1}$ and thus much lower (1.7 % of the median iodine concentration in solid soil) than in the overlaying soils. Therefore, we concluded that precipitation was the main iodine source. Canopy throughfall exhibited an iodine concentration of 1.42 µg L$^{-1}$ including iodine washed off from leaves which was in the lower range of global iodine concentrations in rainwater (0.5-5.0 µg L$^{-1}$; Whitehead, 1984). Combining these values with the high annual rainfall in the ReBAMB of around 2800 mm leads to an annual iodine deposition 3.98 mg m$^{-2}$.

In seven out of nine soil profiles (except for profiles L4: 5 cm and R4: 5 cm) the highest iodine concentration was found at ≥15 cm (L1: 20 cm, L2: 20 cm, L3: 15 cm, R1: 40 cm, R2: 15 cm, R3: 20 cm, R5: 30 cm) indicating that the atmospheric derived iodine is transported to and retained in deeper soil horizons (Figure 3). The Br concentrations in solid samples were, with 43.7 mg kg$^{-1}$ ((R2: 65 cm) and 165.3 mg kg$^{-1}$ (L2: 20 cm)), comparable to those of iodine, and showed a similar maximum at the same depth (L1: 20 cm, L2: 20 cm, L3: 15 cm, R1: 40 cm, R2: 15 cm, R3: 20 cm, R5: 30 cm). This relationship is also expressed by a relative constant iodine to Br-ratio (I:Br) of about 1 throughout the profile (range: 0.77-1.24; median: 0.97) with slightly higher values in deeper horizons (median 5 cm: 0.9, 15-20 cm: 0.95, ≥ 30 cm: 1.14). The Br concentration in throughfall was more than 3.5 times higher than that of iodine, resulting in a smaller I:Br-ratio of 0.28. The I:Br in seawater was found to be 0.001 (Winchester and Duce, 1967). Consequently, much more Br than iodine entered the soils by atmospheric deposition. In the soil, a nearly five times higher I:Br ratio indicates a higher mobility of Br than iodine.



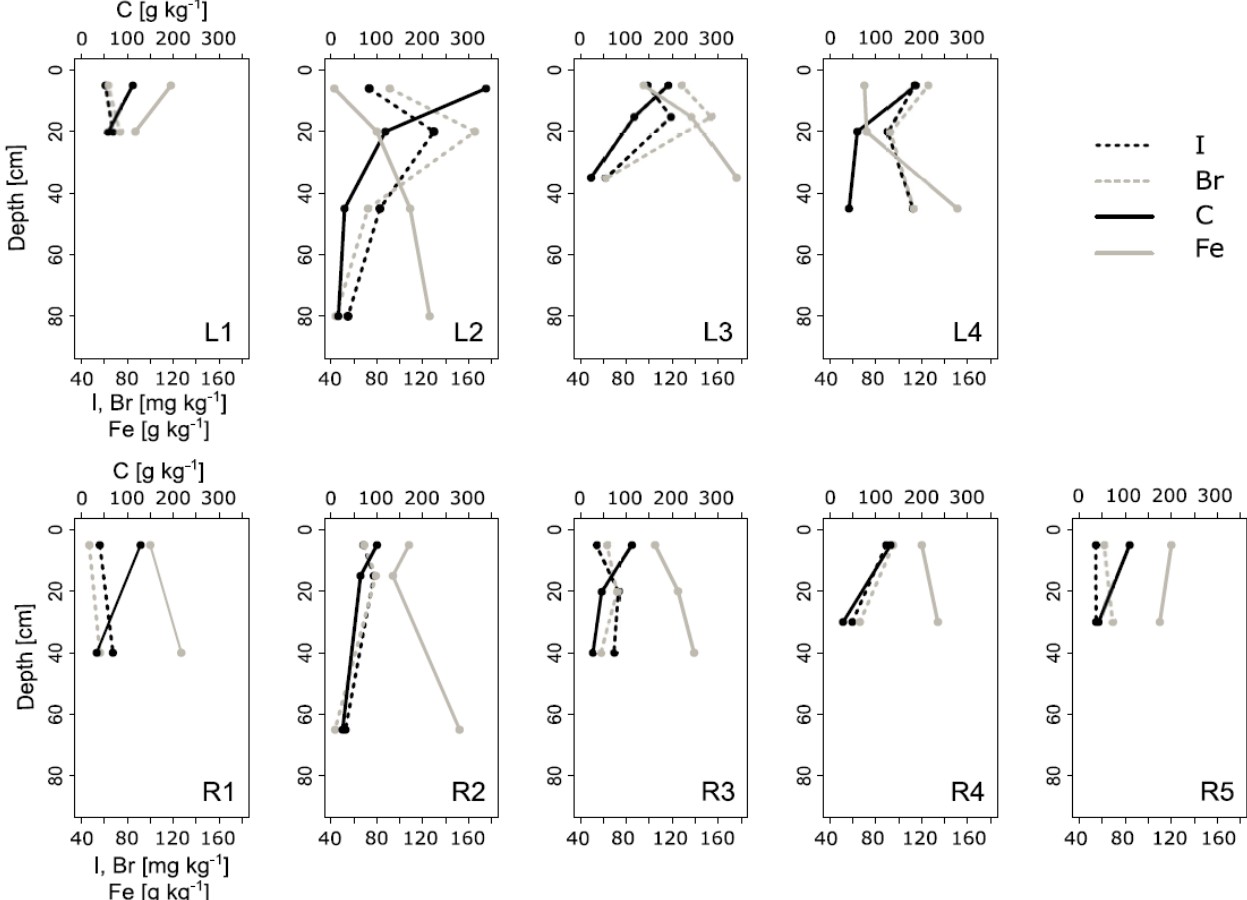

**Figure 3: Concentrations of iodine (I), bromine (Br) [mg kg$^{-1}$], carbon (C) and iron (Fe) [g kg$^{-1}$] in solid soil samples of profiles L1-L4 and R1-R5, shown for the average depth for each sampled horizon.**

**3.2 Solid phase iodine binding**

5    According to previous studies, OM and Fe-oxides are the most important components for solid phase iodine binding in soils. In the ReBAMB soils, maxima of organic C concentrations were found in topsoil horizons (5-6 cm) with concentrations of up to 338 g kg$^{-1}$ (L2 Ah, 6 cm). Lowest C concentrations of 16.3 g kg$^{-1}$ (L2 Bw2, 80 cm) appeared below 30 cm depth (Figure 3). The Fe concentrations increased with depth in all soil profiles showing values of 42.7 g kg$^{-1}$ (L2 Ah, 6 cm) in the topsoil horizons and up to 175.8 g kg$^{-1}$ (L3 Bw, 35 cm; Figure 3) in subsoil horizons, except for the profiles L1 and R5, where Fe

10   concentrations were slightly higher in topsoil horizons (5 cm). However, the profiles L1 and R5 were shallow and might show higher Fe concentrations with greater depths. Thus, these soils provided a high IFP in topsoils (0-5 cm) related to the high OM concentrations and in depths below 30 cm related to the high Fe content.





The solid phase SE analyses showed that between 48 and 152 % (median: 78.7 %) of total iodine (Figure 4) but only between 12.3 % and 72.1 % (median: 30.8 %) of total Br in the soils was associated with reducible components (F4; iron sesquioxides ($Fe_2O_3$), hydroxides ($Fe(OH)_3$) and oxide hydroxides (FeO(OH)). The percentage (fraction) of iodine-Fe-complexes increased with soil depth (except for R3), from 70 % in the upper 5 cm to 82 % at 15-20 cm depth and to 107 % at 30-40 cm depth

5 (percentage as median per depth-range). The highest absolute concentration of iodine bound to Fe-oxides phases (F4) was found in the accumulation horizons of total iodine (≥15 cm). This was attributed to higher Fe contents in subsoil horizons, presumably in the form of Fe-oxides, derived from co-translocation of Fe with clay particles and dissolved organic matter by eluviation-illuviation processes (secondary accumulation) (Maniyunda et al., 2015). The median I:Br-ratio in the F4-leachates was 2.5 (median 5 cm: 3.5, 15-20 cm: 1.8, ≥ 30 cm: 2.4) indicating that iodine was associated with Fe-oxides to a higher extent

10 than Br in all horizons, but most pronounced in topsoil horizons. Despite this, the correlation between iodine and Fe in the solid phase was statistically not significant (r = -0.31).

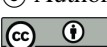

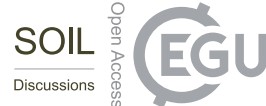

**Figure 4: Left: Extractable iodine in percentage of total iodine amounts in the respective soil horizon [%] of each extraction step F1-F5 in profiles L1 (a), L3 (b), R1 (c), R3 (d), R5 (e). Right: Total iodine concentration in soil profiles [mg kg$^{-1}$].**





The indirect determination of Org-I (F5) by means of the solid phase SE revealed that between 0 - 41 % (median: 10.3 %) of iodine and between 0 and 60.7 (median: 25 %) of Br was associated with soil OM. The largest fractions of Org-I (F5) were found in the topsoils and declined with soil depth (except for R3) indicating that formation of Org-I takes place in the upper soil horizon. However, considerable amounts of iodine in rain might already exist as Org-I (Gilfedder et al., 2010). The correlations of iodine with C ($r_S$= 0.42, p= <0.05) and Br with C ($r_S$= 0.57, p= <0.01) suggest a weak association with C. A reason for the weak correlation between iodine and C in solid samples might be the relatively high C content (median: 62 g kg$^{-1}$) in relation to the iodine and Br content (median: 69 mg kg$^{-1}$, 71 mg kg$^{-1}$), respectively. Thus, the amount of OM was not a limiting factor for iodine and Br fixation. Hence, increasing C concentration did not lead to a stronger selective iodine fixation. The slightly higher correlation of Br with C is also expressed in the low I:Br-ratio of F5 (median: 0.29), with the highest ratio in topsoils (median 5 cm: 0.62, 15-20 cm: 0.17, ≥ 30 cm: 0.30). Thus, more Br was associated to OM than iodine, particularly below topsoils.

The extraction steps F1 (median: iodine: 0.2 %, Br: 0.7 %), F2 (median: iodine: 1.9 %, Br: 3.3 %) and F3 (median: iodine: 12.1 %, Br: 26.7 %) together did release less than 20 % of total iodine, but 69 % of Br per sample (median: 14.2 %) Accordingly, I:Br-ratios were low in F1 (median:0.28), F2 (median: 0.48) and F3 (median:0.36) indicating weaker retention of Br. In four out of twelve samples the determined sum of the iodine content extracted during F1-F4 were higher (1.4 %, 5.8 %,47.7 % and 69.7 %) than the values of the total iodine content caused by cumulated uncertainties of consecutive extractions and inhomogeneity of the soil sample. The SE results showed that Br is more mobile and bound weaker to the soil matrix than iodine (F1-F3). It seemed that Fe-oxides stabilize and retain more iodine than Br (F4), but OM stabilized more Br than iodine (F5) at all depths.

### 3.3 Iodine, Br and DOC in water leachates

Iodine and Br concentrations in soil water leachates varied between 0.2 and 21.9 µg L$^{-1}$ (median: 0.9 µg L$^{-1}$) and 0.37 and 47 µg L$^{-1}$ (median: 3.2 µg L$^{-1}$), respectively. This results in a median I:Br-ratio of 0.31 in leachates with the highest I:Br-ratio at depths ≥ 30 cm (median 5 cm: 0.29, 15-20 cm: 0.18, ≥ 30 cm: 0.83) confirming the SE results. The DOC-concentrations in leachates ranged between 0.8 and 17.7 mg L$^{-1}$ (median: 2 mg L$^{-1}$). Iodine, Br and DOC showed generally low mobility (median: 0.01 %, 0.04 % and 0.03 %, respectively). The maximum concentrations of iodine, Br and DOC were found in the leachates of the topsoil horizons and decreased with soil depth, except for profile R2, where the highest iodine mobility was found at 65 cm (Figure 5). Iodine, Br and DOC were significantly correlated in the leachates (iodine-DOC: 0.7, p= <0.001, Br-DOC 0.74, p= <0.001) suggesting that water soluble iodine and Br at least partly existed as dissolved organo-halogen compounds.





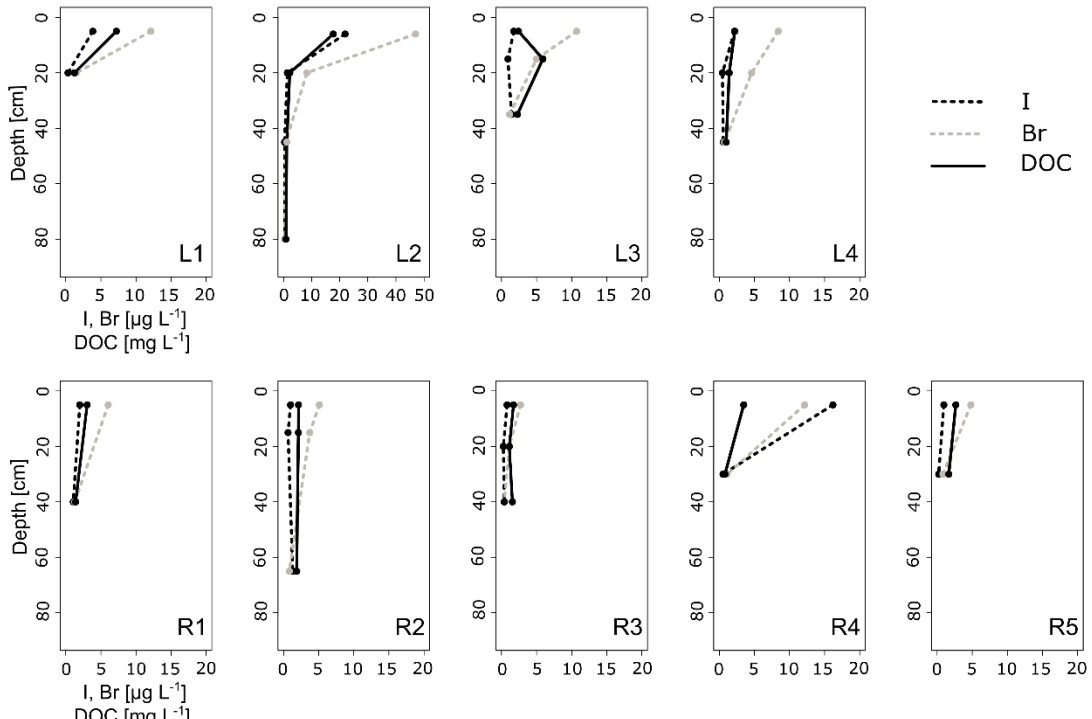

**Figure 5: Iodine (I), bromine (Br) and dissolved organic carbon (DOC) concentrations in leachates [µg L⁻¹] of profiles L1-L4 and R1-R5, shown for the average depth for each sampled horizon, with x-axis scale set to 50 for L2.**

According to the results obtained by solid phase SE, the Fe-oxides provided the major mineral surface for iodine binding and

retention but not for Br. We assume that iodine and Br are transported as DOC-complexes from topsoil to deeper soil horizons, but Br was presumably also transported as bromide ($Br^-$). The oxidation potential of iodine is lower (lower electronegativity) than that of Br, and iodine is more easily oxidized to hypoiodous acid (HOI) than Br to hypobromous acid (HOBr) which can only be oxidized enzymatically by haloperoxidase (Martínez-Cortizas et al., 2016; Li et al., 2011). The latter generally results in lower organo-Br formation in soils (Gilfedder et al., 2011). Once in the HOI/ HOBr-form, iodine and Br incorporate quickly

into OM via covalent bindings to C. Accordingly, highly mobile and rather conservative inorganic species of Br, bromide, are more likely to occur in soils and aqueous systems than inorganic species of iodine. Thus, Br can be easily mobilised as bromide (Gilfedder et al., 2011) and is therefore more mobile than iodine. In deeper soil horizons, the DOC-iodine (DOC-I) complexes can be adsorbed on Fe-oxide surfaces. We assumed that the amount of iodine bound to Fe-oxides as revealed from SE analyses was eventually overestimated. Due to the high affinity of iodine binding to DOC it is likely that most iodine is bound to DOC

and the dissolution of Fe-oxides during step F4 released DOC-bound iodine to Fe-oxides as found by Li et al. (2013). Thus, the fraction of iodine bound to OM in deeper soil horizons was likely underestimated.





The increasing Fe concentrations with soil depth indicated increasing iodine retention capacity through adsorption of DOC-I complexes on Fe-oxide surfaces. This caused the low amounts of leachable iodine and DOC in these horizons. The Fe-Oxides were stable under oxic conditions but will be dissolved under reducing conditions (e.g. if water logged) releasing the adsorbed DOC-I complexes. However, Fe and iodine mobilization in ReBAMB soils caused by reducing conditions is unlikely to occur

because of the high infiltration rates (> 100 mm h$^{-1}$) and steep slopes that quickly and continuously transfer water towards the next stream (Dehaspe et al., 2018; Solano-Rivera et al., 2019). Moreover, the low mobility of iodine as DOC-I-Fe-oxide-complex was caused by the fact that Fe-oxides protect OM against degradation. This promotes effective OM stabilisation, increasing with soil depth (Kaiser et al., 1996; Hagedorn et al., 2015). Thus, stabilized DOC-I-Fe-oxide complexes may become enriched over longer time periods resulting in low DOC release (Aran et al., 2001; Camino-Serrano et al., 2014). The

Iodine accumulation in acidic (pH < 5) tropical soils is strongly connected to the distribution and abundance of Fe-oxides and the related stabilization of OM. Both are mainly controlled by weathering and pedogenic processes. The related high enrichment and low mobility of iodine in these tropical soils is therefore mainly a result of their high age.

### 3.4 Iodine in stream water

The low mobility and high retention of iodine in the ReBAMB soils led to iodine concentrations in the San Lorencito stream

water of 0.9-1.3 µg L$^{-1}$ (median: 1.0 µg L$^{-1}$), despite higher iodine concentration in throughfall (1.42 µg L$^{-1}$). Iodine concentrations in rivers and lakes found by Gilfedder et al. (2010) in temperate regions ranged between <1 and 5.1 µg L$^{-1}$. In contrast, the concentration of Br in throughfall (5.0 µg L$^{-1}$) was lower than in stream water (mean: 6.7 µg L$^{-1}$). The significant correlation of Br concentration in stream water and discharge ($r_s$= 0.61, p<0.05) suggested that the soils retain incoming Br by a far lesser extent. In contrast, no correlation was found between iodine concentrations and discharge in stream water. The

high affinity of iodine to bind to OM and subsequent retention of DOC-I by Fe-oxides led to a filter effect of iodine in the subsoils.

### 4 Conclusion

Iodine in the San Lorencito stream water catchment is mostly derived from the atmosphere and is retained in the soil. We found the ReBAMB soils to be highly enriched in iodine. Iodine mobility is generally low (0.01 % of total iodine), though

higher in topsoils (5-6 cm), declining with increasing soil depth. The transport of iodine appears to be related to DOC and iodine is retained in deeper soil horizons by Fe-oxides as DOC-I complexes. Higher iodine concentrations in throughfall compared to stream concentrations show that tropical soils are an iodine sink, while Br is retained by a far lesser extent. Heavy rainfall events have only little influence on iodine mobilization. Since incoming iodine is stored and not mobilized, it can be concluded that the studied ecosystem is not saturated with iodine and exhibits high iodine retention capacity, which results in

low iodine in aquatic systems and low bioavailability.



**Author contribution**

HB, LB and KS designed the experiments and carried them out. CB organized the field campaign. LB prepared the manuscript with contributions from all co-authors.

**Competing interests**

The authors declare that they have no conflict of interest.

**Acknowledgements**:

The authors would like to thank the staff of the ReBAMB research station for their support, A. Calean and P. Schmidt for
technical assistance and the (anonymous) reviewers for their helpful input

This study was partially supported by the Deutsche Akademische Austauschdienst (DAAD) grant Promos.

CB acknowledges support by the UCR research council project B8709.

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
