# Peer review of "Supporting information to: Iron oxides control sorption and mobilisation of iodine in a tropical rainforest catchment"

_SOIL, 2020_

## Referee Comment (RC1) · Anonymous Referee #1 · 30 Apr 2020

General comments This study deals with distribution of iodine in soils in tropical rainforest. Iodine in soils from nine profiles are investigated by sequential extraction technique. The observed data for iodine in tropical rainforest is valuable. The study is well-organized and the manuscript is well-written. But the considerable revisions for following points should be needed before acceptance.

1. My major concern is that interruption of results from the sequential extraction may have made erroneous conclusion. Authors concluded that iron oxide is the main factor controlling iodine mobility in soil by the fact that major fraction of iodine was F4. It is well known that iodine is highly mobilized as iodide under anoxic soil conditions. Extraction step with reducing agent ($NH_2OH$ HCl) can alter iodine form in soil resulting high percentage in F4. This is not necessarily indicating that iodine adsorbed to iron oxide. In

addition, sum of F1 to F4 reached upto about 150% in some soil in Figure 4, suggesting that F4 can be overestimated and F5 can be underestimated in this method. Authors should discuss in detail the potential artifact and defect of the extraction procedure for iodine fraction. 2. Many previous studies indicate that soil organic matter, rather than iron oxide, control the iodine mobility in soil. Author should review the previous knowledge in detail about soil component controlling iodine mobility, and novelty of this study should be discussed. In addition, the manuscript lacks references of recent papers on the topics including dynamics of iodine in forest ecosystem, speciation of iodine in soil and water, and mobility of iodine in soil, for example, Roulier et al (2019) Chemosphere 224, 20; Humphrey et al. (2020) Environ. Sci. Technol. 54, 1443; Takeda et al. (2018) Soil Sci. Soc. Am. J. 82, 815; Unno et al. (2017) J. Environ. Radioactiv. 169–170, 131.

Specific comments Page 3 Line 8 Refer the soil classification system exactly. Page 12 Line 5, and Page 14 Line 17-19 Scatter plot should be given to explain the correlation analysis.

---

## Referee Comment (RC2) · Anonymous Referee #2 · 15 May 2020

General comments

This study concerns the evaluation of the soil-related factors that dominate iodine retention and mobilization in tropical soils and solid phase sequential extraction was used to identify iodine binding forms in soils. It is interesting the evaluation of the mobilization along the nine soil profiles, however the main concern relates to the novelty of this study. Author should review the previous knowledge in detail about soil component controlling iodine mobility, and the novelty of this study should be stressed. The title "Iron oxides control sorption and mobilisation of iodine in a tropical rainforest catchment" presents a problem that is repeated throughout the article. I would expect a focus on the effect of Fe oxides on the mobilization of iodine, but Fe oxides have not been identified in this work.

In its current state, the manuscript is not suitable for publication in SOIL. I hope that the comments below are useful, and look forward to reading more about this work in the future.

Specific comments

Page 1 Lines 27-31: please cite more recent references (Humphrey et al. (2017) Environmental Science Processes and Impacts and references therein).

Page 5 Lines 1-2: the data related to "Physical soil properties for each horizon were recorded in the field including texture, aggregate type, degree of rooting, skeleton and pore volume (see Supporting Information)" are not provided in the Supporting Information.

Page 5 Lines 7-9: "The pH value of the air-dried soil was measured in water with a soil:solution ratio of 1:2.5. The water content ($\theta$) was determined gravimetrically for all horizons of profiles L2 and R2. Stream water temperature, pH, redox potential (Eh) and conductivity (EC) were measured in-situ using a handheld Hanna multi-parameter probe (HI 98195)" Some pH values are missing in tables S1 and S2. In addition, the role of pH on organo-iodine formation and the dissolution of Fe-oxides should be clearly explained. The authors mention redox potential (Eh) and conductivity (EC), however, no such results are presented in the manuscript.

Page 5 Line 18: the sequential extraction procedure according Schmitz and Aumann (1995) is a pretty well-established procedure. The OM-bound iodine can be severely underestimated using this method. Fractionation studies have a limited scope for iodine analysis in soils as operational procedures do not necessarily yield discrete fractions. A much greater emphasis has been placed on chemical speciation analysis and various methods have been successfully developed for analyzing chemical species. In addition, the authors should underline the potential artifacts created using this extraction procedure, and should also consider limitations/problems of the analytical approach. Overall, I recommend adding a paragraph with a list and more detailed discussion

about the uncertainties of the chemical fractionation.

Page 7 Line 25: My main concern related to this work is that it is not enough to identify the major Fe concentration using an energy-dispersive X-ray fluorescence spectrometer. In addition, the method of Cheburkin and Shotyk (1996) analyzed Pb and trace elements in peats. This result should be supported by the analysis of Fe components, e.g., selective extraction methods (for tropical soils see the methods described in Coward et al., 2017 Geoderma) or XRD.

Page 9 Fig. 3, Page 11 Fig. 4, and Page 13 Fig. 5: please report the correlation values.

Page 9 Line 5: more references are needed. Suggested readings: (Schlegel et al., 2006 Geochimica et Cosmochimica Acta; Shetaya et al., 2012 Geochimica et Cosmochimica Acta, Li et al., 2017 Journal of Contaminant Hydrology, Xue et al., 2019 Science of the Total Environment, Qian et al., 2020 Science of the Total Environment).

Page 10 Lines 1-3 and Lines 6-8: "The solid phase SE analyses showed that between 48 and 152 % (median: 78.7 %) of total iodine (Figure 4) but only between 12.3 % and 72.1 % (median: 30.8 %) of total Br in the soils was associated with reducible components (F4; iron sesquioxides (Fe2O3), hydroxides (Fe(OH)3) and oxide hydroxides (FeO(OH))" and "This was attributed to higher Fe contents in subsoil horizons, presumably in the form of Fe-oxides..." again, the solid phase extraction and speciation of Fe oxide would be needed.

Page 11 Fig. 4: the sum of F2, F3, and F4 fractions reaches up to 150%. Please explain why.

The section about conclusions could be improved; it is rather a summary of results than a real presentation of conclusions. What are the new insights into Fe oxides-mediated sorption and mobilization of iodine gained by applying the present fractionation method?

---

## Referee Comment (RC3) · Anonymous Referee #3 · 16 May 2020

This paper examines soil pools and hydrologic fluxes of iodine in a tropical forest, in conjunction with other measurements of soil and water chemistry. I concur with the authors' overall interpretation of the data, and I think that the data presented here are potentially useful for the biogeosciences / soil science community, as iodine remains an understudied element.

However, I have some significant concerns about details of data interpretation, specifically in the context of the soil extractions, but believe that these can be remedied in a revision. Specifically, the extractions used here cannot discern whether iodine was in fact directly associated with organic matter vs iron. The hydroxylamine extraction only reduces a fraction of the short-range-ordered Fe phases, a fact that is well established in the literature. To extract crystalline Fe, which is likely to be abundant in these soils

as demonstrated by the high total Fe content of the subsoil, at least one (if not multiple) extractions with dithionite would be needed. Therefore, attribution of "residual" iodine following extraction by hydroxylamine to an organic-bound (as opposed to Fe-bound) iodine pool is erroneous. Second, even the iodine released by hydroxylamine may have been proximately bound by organic matter (as suggested by the authors themselves in the discussion). This needs to be made clear earlier in the paper.

Overall, interpretation of what these extractions mean needs to be more precise and cautious. However, the fact that most of the iodine was typically released in the hydroxylamine extraction is interesting and important and suggests that the iodine was associated with SRO Fe phases and/or organic matter bound with these phases. If the DOC content of the hydroxylamine extraction was measured, the authors could assess whether DOC was adsorbed or coprecipitated using the molar DOC/Fe ratios.

Second, the rationale for sampling the different soil profiles to different depths was not at all clear. Given that bedrock was not present, why were some profiles only sampled to very shallow depths? This complicates our interpretation of the data. It is difficult to interpret a depth profile of only two measurements in terms of transport/retention dynamics of an element. Some of the "A" horizons are clearly "O" horizons based on high concentrations of C.

Overall, the writing was not well-focused and often rambled. For example, the first two paragraphs of page 2 need clear topic sentences and organization to guide the reader through an argument. The rationale for studying I and Br together should be introduced, since these trends become a major part of the results.

In the introduction, it would be helpful to briefly describe the biogeochemistry of iodine in a bit more detail, because this is an uncommon topic in the literature. Some of these topics are addressed in the discussion. (e.g. differences in binding of iodine and bromine). For example, what are the major species of iodine in soil? What kinds of bonds do they form with SOM and the major mineral phases?

Specific comments: L17: "Stream water was sampled randomly over a period of five weeks": this is haphazard, not random sampling I believe? P1 Introduction: discussion of the health impacts of iodine seems remotely linked to the focus of this paper; better to make this explicit or remove P2 L22: Avoid overgeneralizing about "tropical soils". Note that almost all of the global soil orders (except Gellisols) can be found in the tropics. Clarify the scope of your study accordingly. P3: "(REF to the classification system)" – revise or delete Given your previous description, it seems as if this soil is really a Ferralsol? P4: Why did your sampling depth vary among soil pits (e.g. 0.5 m or 1 m) P5: The water sampling scheme is unclear. Did your sampling span both base flow and stormflow conditions? P6: Note that only a fraction of soil Fe phases are reduced with hydroxylamine. Dithionite is needed to reduce crystalline Fe. Furthermore, the hydroxylamine extraction will also extract iodine bound with organic matter, because Fe-associated organic matter is released. Therefore, the F5 fraction cannot be used to represent OM-bound iodine, as substantial iodine may remain associated with Fe (and other mineral) phases, and previous extractants (e.g. F4) likely included OM-bound iodine. See for example Coward et al. 2017 10.1016/j.geoderma.2017.07.026 Section 2.3.2: what kind of water was used for the leaching experiments? P8 L8: What kind of "temperate soils", and how representative are these? Note the tremendous diversity in soil types and likely iodine input/output budgets among ecoystems. P9 L5: What "previous studies"? Present your data first. You need to cite specific literature if you want to compare. P9 L7: This organic C concentration is too high for a mineral A horizon. This indicates that an O (organic) horizon was sampled. P10: Note that the Fe-associated iodine is underestimated because your extractions did not release crystalline Fe phases, which likely dominated here (especially in the subsoil) Figure 4: It is concerning that three samples do not have any F5 fraction (difference between total and extracted iodine). In how many cases was this value negative (e.g. more iodine was extracted than in the total measured sample) P12 L1: Following the reasoning above, you cannot determine that F5 fraction iodine is associated with SOM. Revise. P14 L1-5: The occurrence of Fe reduction and DOC/nutrient mobilization in tropical

soils has received significant recent attention, even (and especially) in systems with high rainfall and high infiltration rates. Iron reduction is widespread in these kinds of ecosystems. It would help to read and cite relevant literature here. P15 L6 "Moreover, the low mobility of iodine as DOC-I-Fe-oxide- complex was caused by the fact that Fe-oxides protect OM against degradation." Note recent findings that challenge this notion; Fe/C/nutrient interactions can be dynamic.

―――――――――――――――――――――

---

## Author Comment (AC1) · 23 Jun 2020

We would like to thank reviewer 1 for the helpful comments and suggestions to improve the quality of our work. Detailed responses to the comments of R1 are given below. The original comments by R1 are between quotation marks.

"General comments This study deals with distribution of iodine in soils in tropical rainforest. Iodine in soils from nine profiles are investigated by sequential extraction technique. The observed data for iodine in tropical rainforest is valuable. The study is well-organized and the manuscript is well-written. But the considerable revisions for following points should be needed before acceptance."

1. "My major concern is that interruption of results from the sequential extraction may

have made erroneous conclusion. Authors concluded that iron oxide is the main factor controlling iodine mobility in soil by the fact that major fraction of iodine was F4. It is well known that iodine is highly mobilized as iodide under anoxic soil conditions. Extraction step with reducing agent (NH2OH HCl) can alter iodine form in soil resulting high percentage in F4. This is not necessarily indicating that iodine adsorbed to iron oxide."

It would be possible that NH2OH HCl reduces some iodine to iodide under the anoxic conditions in step F4. But this requires the extracted (and reduced) iodine be either sorbed to Fe phases or OM. We are not excluding, that iodine bound to DOC was extracted during F4, when Fe-oxide became dissolved. As described in the manuscript, we believe that organo-iodine is the preliminary form of iodine (DOM-I), which is then bound to Fe-oxides in the deeper soil layers. Several studies have already shown, that the main form in the soils is organic iodine (Unno et al., 2017; Xu et al., 2011) as inorganic iodine is transformed rapidly to organic forms in the soil surface (Takeda et al., 2015; Hu et al., 2012) and iodine is entering the soil to a large extent as organic iodine.

2. "In addition, sum of F1 to F4 reached up to about 150% in some soil in Figure 4, suggesting that F4 can be overestimated and F5 can be underestimated in this method. Authors should discuss in detail the potential artifact and defect of the extraction procedure for iodine fraction."

In principal, all sequential extraction procedures include (large) uncertainties arising from re-adsorption to the residue of the extraction step, cross-contamination, incomplete digestion, release of other iodine forms, volatilization or transformation of iodine, especially in a strong acid/base solution (Shimamoto et al., 2011; Hou et al., 2009). We have already mentioned that in the manuscript "In four out of twelve samples the determined sum of the iodine content extracted during F1-F4 were higher (1.4 %,5.8 %,47.7 % and 69.7 %) than the values of the total iodine content caused by cumulated uncertainties of consecutive extractions and inhomogeneity of the soil sample." (Page

12 Lines 15-17) This probably means that Iodine dissolved during the first, second and third extractions may be re-adsorbed on the active phases/residues of F3 leading to an overestimating of iodine bound to iron oxides. However, from our data we can conclude that only a small amount of iodine is water leachable due to our low iodine concentration in the river water during base and stormflow conditions. Iodine extracted during F4 may also be reabsorbed on the remaining phase (organic matter), which would lead to an underestimation of F4. This would support our findings that most of the iodine is associated to Fe-oxides.

As mentioned in the manuscript, it is likely that due to the high affinity of iodine binding to OM most of the iodine in the soil is bound to OM and the dissolution of Fe-oxides during step F4 released DOM-bound iodine to Fe-oxides similar as found by Li et al. (2013). Thus, the fraction of iodine bound to OM in deeper soil horizons was likely underestimated. (Page 13 Lines 14-16). Despite all the uncertainties most of the iodine was extracted during the hydroxylamine extraction suggesting that most of the iodine in our soils is associated to SRO Fe phases and/ or OM bound to it as also stated by the third reviewer. We also believe that the exact separation between Fe-oxide bound and organically bound iodine is not the essential point as these components never exist completely separated in soils. The message of our study is that DOM-bound iodine is retained by the high abundance of Fe-oxides in tropical soils leading to enrichment and reduction of iodine release to the adjacent aquatic system. We will try to make this point clearer in the revised manuscript.

3. "Many previous studies indicate that soil organic matter, rather than iron oxide, control the iodine mobility in soil. "

We agree with the reviewer as this is one of our findings. We are also saying that organic matter controls iodine mobility in our soils as iodine is initially transported as DOM-iodine-complex from the upper organic rich soil layer but is retained by Fe-oxides in deeper soil horizons.

4. "Author should review the previous knowledge in detail about soil component controlling iodine mobility, and novelty of this study should be discussed. "

We are aware, that several previous papers have shown that iodine mobility is mainly controlled by organic matter and iodine is leached out as organic iodine from soils (e.g Roulier et al., 2019; Xu et al., 2011; Unno et al., 2017). Our study shows that iodine is transported as DOC-complexes from topsoil to deeper soil horizons (page 13 Line: 5), where the DOC-iodine (DOC-I) complexes are retained through binding to Fe-oxide surfaces (page 13 Lines: 12-13). The high Fe concentrations in our soils cause the low amounts of leachable iodine (and DOC). Due to the high age of our soils and the long exposure time to iodine depositions the soils had a long time to accumulate iodine in the soil during soil formation due to the process of DOC-iodine leaching from topsoil to subsoil and fixation by Fe-oxides. (page 14 Lines: 11-12) Regarding the novelty of this study, we believe that the combination of solid phase iodine binding analyses and mobilisation tests and especially the monitoring of iodine in adjacent aquatic system is novel and indicates the consequences of long term enrichment and retention of iodine in Fe-rich tropical soils for aquatic systems and its potential bioavailability there. We recognized that this has not become entirely clear, and changed the title and put more emphasis on these novel findings in the revised manuscript.

5. "In addition, the manuscript lacks references of recent papers on the topics including dynamics of iodine in forest ecosystem, speciation of iodine in soil and water, and mobility of iodine in soil, for example, Roulier et al (2019) Chemosphere 224, 20; Humphrey et al. (2020) Environ. Sci. Technol. 54, 1443; Takeda et al. (2018) Soil Sci. Soc. Am. J. 82, 815; Unno et al. (2017) J. Environ. Radioactiv. 169–170, 131. Specific comments Page 3 Line 8 "

We will include the references in the revised manuscript.

6. "Refer the soil classification system exactly."

The soils were classified using the World Reference Base for Soil Resources (IUSS

Working Group WRB, 2015). We will include the reference.

7. "Page 12 Line 5, and Page 14 Line 17-19 Scatter plot should be given to explain the correlation analysis."

Will be included in the revised manuscript.

References Hou, X., Hansen, V., Aldahan, A., Possnert, G., Lind, O. C., and Lujaniene, G.: A review on speciation of iodine-129 in the environmental and biological samples, Anal Chim Acta, 632, 181–196, https://doi.org/10.1016/j.aca.2008.11.013, 2009.

Hu, Q. H., Moran, J. E., and Gan, J. Y.: Sorption, degradation, and transport of methyl iodide and other iodine species in geologic media, Appl Geochem, 27, 774–781, https://doi.org/10.1016/j.apgeochem.2011.12.022, 2012.

IUSS Working Group WRB: World reference base for soil resources 2014, update 2015: International soil classification system for naming soils and creating legends for soil maps, World soil resources reports, 106, FAO, Rome, 181 pp., 2015.

Li, J., Wang, Y., Xie, X., Zhang, L., and Guo, W.: Hydrogeochemistry of high iodine groundwater: A case study at the Datong Basin, northern China, Environ. Sci.: Process. Impacts, 15, 848–859, https://doi.org/10.1039/C3EM30841C, 2013.

Roulier, M., Coppin, F., Bueno, M., Nicolas, M., Thiry, Y., Della Vedova, C., Février, L., Pannier, F., and Le Hécho, I.: Iodine budget in forest soils: Influence of environmental conditions and soil physicochemical properties, Chemosphere, 224, 20–28, https://doi.org/10.1016/j.chemosphere.2019.02.060, 2019.

Shimamoto, Y. S., Takahashi, Y., and Terada, Y.: Formation of organic iodine supplied as iodide in a soil-water system in Chiba, Japan, Environ. Sci. Technol., 45, 2086–2092, https://doi.org/10.1021/es1032162, 2011.

Takeda, A., Tsukada, H., Takahashi, M., Takaku, Y., and Hisamatsu, S.: Changes in the chemical form of exogenous iodine in forest soils and their extracts, Radiation

protection dosimetry, 167, 181–186, https://doi.org/10.1093/rpd/ncv240, 2015.

Unno, Y., Tsukada, H., Takeda, A., Takaku, Y., and Hisamatsu, S.'i.: Soil-soil solution distribution coefficient of soil organic matter is a key factor for that of radioiodide in surface and subsurface soils, J Environ Radioactiv, 169-170, 131–136, https://doi.org/10.1016/j.jenvrad.2017.01.016, 2017.

Xu, C., Miller, E. J., Zhang, S., Li, H.-P., Ho, Y.-F., Schwehr, K. A., Kaplan, D. I., Otosaka, S., Roberts, K. A., Brinkmeyer, R., Yeager, C. M., and Santschi, P. H.: Sequestration and remobilization of radioiodine (129I) by soil organic matter and possible consequences of the remedial action at Savannah River Site, Environ. Sci. Technol., 45, 9975–9983, https://doi.org/10.1021/es201343d, 2011.

––––––––––––––––––––––––––––

---

## Author Comment (AC2) · 23 Jun 2020

We would like to thank reviewer 2 for the detailed comments and suggestions to improve the quality of our work. Detailed responses to the comments of R2 are given below. The original comments by R2 are between quotation marks.

"General comments" 1. "This study concerns the evaluation of the soil-related factors that dominate iodine retention and mobilization in tropical soils and solid phase sequential extraction was used to identify iodine binding forms in soils. It is interesting the evaluation of the mobilization along the nine soil profiles, however the main concern relates to the novelty of this study. Author should review the previous knowledge in detail about soil component controlling iodine mobility, and the novelty of this study

should be stressed."

Regarding the novelty of this study, we believe that the combination of solid phase iodine binding analyses and mobilisation tests and especially the monitoring of iodine in adjacent aquatic system is novel and indicates the consequences of long-term enrichment and retention of iodine in Fe-rich tropical soils for aquatic systems and its potential bioavailability there. We recognized that this has not become entirely clear, and changed the title and put more emphasis on these novel findings in the revised manuscript. We are aware, that several previous papers have shown that iodine mobility is mainly controlled by organic matter and iodine is leached out as organic iodine from soils (e.g Roulier et al., 2019; Xu et al., 2011a; Unno et al., 2017). Our study shows that iodine is transported as DOC-complexes from topsoil to deeper soil horizons (page 13 Line: 5), where the DOC-iodine (DOC-I) complexes are retained through binding to Fe-oxide surfaces (page 13 Lines: 12-13). The high Fe concentrations in our soils cause the low amounts of leachable iodine (and DOC). Due to the high age of our soils and the long exposure time to iodine depositions the soils had a long time to accumulate iodine in the soil during soil formation due to the process of DOC-iodine leaching from topsoil to subsoil and fixation by Fe-oxides. (page 14 Lines: 11-12)

2. "The title "Iron oxides control sorption and mobilisation of iodine in a tropical rainforest catchment" presents a problem that is repeated throughout the article."

Based on this comment we have decided to change the title of the manuscript to "Organo-iodine sorption to iron-oxides controls high enrichment and low mobility of iodine in soils of a pristine tropical rainforest".

3. "I would expect a focus on the effect of Fe oxides on the mobilization of iodine, but Fe oxides have not been identified in this work."

As mentioned above previous studies showed that organic matter is a main factor controlling iodine mobility in surface soils. But we showed that Fe-oxides stabilize organically bound iodine by strong sorption in the soils and protect it against mobilisation. The

exact identification of the Fe-oxides was not part of the study and would not change the fact that some Fe phases act as the main sorbent for (organo-) iodine.

"In its current state, the manuscript is not suitable for publication in SOIL. I hope that the comments below are useful, and look forward to reading more about this work in the future." We thank the Reviewer for his/her constructive criticism and hope to provide an acceptable revision for publication in SOIL.

"Specific comments"

4. "Page 1 Lines 27-31: please cite more recent references (Humphrey et al. (2017) Environmental Science Processes and Impacts and references therein). "

We will include the references.

5. "Page 5 Lines 1-2: the data related to "Physical soil properties for each horizon were recorded in the field including texture, aggregate type, degree of rooting, skeleton and pore volume (see Supporting Information)" are not provided in the Supporting Information. "

We will update the SI.

6. "Page 5 Lines 7-9: "The pH value of the air-dried soil was measured in water with a soil:solution ratio of 1:2.5. The water content () was determined gravimetrically for all horizons of profiles L2 and R2. Stream water temperature, pH, redox potential (Eh) and conductivity (EC) were measured in-situ using a handheld Hanna multi-parameter probe (HI 98195)"

a. "Some pH values are missing in tables S1 and S2. " The pH was only measured in selected profiles (L2, R2 and R4) because the sample amount was limited. But it can be assumed that they would be in the same range of 4.1-4.9

b. "In addition, the role of pH on organo-iodine formation and the dissolution of Fe-oxides should be clearly explained." We agree that the pH can have an impact on

iodine speciation and mobility. But a detailed discussion on the effect of the pH on iodine speciation was not part of the manuscript and can be found elsewhere (e.g. (Schwehr et al., 2009; Kaplan, 2003; Xu et al., 2011b; Yeager et al., 2017; Sheppard et al., 1995; Yoshida et al., 1992)).

c. "The authors mention redox potential (Eh) and conductivity (EC), however, no such results are presented in the manuscript. " Will be included in the revised manuscript.

7. "Page 5 Line 18: the sequential extraction procedure according Schmitz and Aumann (1995) is a pretty well-established procedure. The OM-bound iodine can be severely underestimated using this method. Fractionation studies have a limited scope for iodine analysis in soils as operational procedures do not necessarily yield discrete fractions. A much greater emphasis has been placed on chemical speciation analysis and various methods have been successfully developed for analyzing chemical species. In addition, the authors should underline the potential artifacts created using this extraction procedure, and should also consider limitations/problems of the analytical approach. Overall, I recommend adding a paragraph with a list and more detailed discussion about the uncertainties of the chemical fractionation. "

In Principal all sequential extraction procedures includes (large) uncertainties arising from re-adsorption to the residue of the extraction step, cross-contamination, incomplete digestion, release of other iodine forms, volatilization or transformation of I, especially in a strong acid/base solution (Shimamoto et al., 2011; Hou et al., 2009). We have already mentioned that in the Manuscript "In four out of twelve samples the determined sum of the iodine content extracted during F1-F4 were higher (1.4 %,5.8 %, 47.7 % and 69.7 %) than the values of the total iodine content caused by cumulated uncertainties of consecutive extractions and inhomogeneity of the soil sample." (Page 12 Lines 15-17). This probably means that Iodine dissolved during the first, second and third extractions may be re-adsorbed on the active phases/residues of F3 leading to an overestimating of iodine bound to iron oxides. However, from our data we can conclude that only a small amount of iodine is water leachable due to our low iodine

concentration in the river water during base and stormflow conditions. Iodine extracted during F4 may also be reabsorbed on the remaining phase (organic matter), which would lead to an underestimation of F4. This would support our findings that most of the iodine is associated to Fe-oxides. As written in the manuscript, it is likely that due to the high affinity of iodine binding to OM most of the iodine in the soil is bound to OM and the dissolution of Fe-oxides during step F4 released DOM-bound iodine to Fe-oxides similar as found by Li et al. (2013). Thus, the fraction of iodine bound to OM in deeper soil horizons was likely underestimated. (Page 13 Lines 14-16). Despite all the uncertainties most of the iodine was extracted during the hydroxylamine extraction suggesting that most of the iodine in our soils is associated to SRO Fe phases and/ or OM bound to it as also stated by the third reviewer. We also believe that the exact separation between Fe-oxide bound and organically bound iodine is not the essential point as these components never exist completely separated in soils. We try to show that the high retention of iodine through adsorption of DOM-iodine complexes to Fe-oxides is the major process of iodine enrichment in tropical soils and the resulting low iodine concentrations in adjacent drainage systems. This is to our knowledge the novelty in this study. We will make this point clearer in the manuscript.

8. "Page 7 Line 25: My main concern related to this work is that it is not enough to identify the major Fe concentration using an energy-dispersive X-ray fluorescence spectrometer. In addition, the method of Cheburkin and Shotyk (1996) analyzed Pb and trace elements in peats. This result should be supported by the analysis of Fe components, e.g., selective extraction methods (for tropical soils see the methods described in Coward et al., 2017 Geoderma) or XRD. "

This method is unfortunately not available in our laboratory. However, the identification of the Fe phases was not part of the study and would not change the fact that most of the iodine is associated with Fe-oxides.

9. "Page 9 Fig. 3, Page 11 Fig. 4, and Page 13 Fig. 5: please report the correlation values." We have stated the correlation of total iodine with total carbon in the soil on

page page 12 Lines 4-5: "The correlations of iodine with C (rS= 0.42, p= <0.05) and Br with C (rS= 0.57, p= <0.01) suggest a weak association with C" and the correlation of iodine with Fe on page 10 Line 10-11: "Despite this, the correlation between iodine and Fe in the solid phase was statistically not significant (r = -0.31)". We will add a scatter plot with this correlation. We have shown the correlation for iodine and DOC in the leachates on Page 12 Lines 27-28: "Iodine, Br and DOC were significantly correlated in the leachates (iodine-DOC: 0.7, p= <0.001, Br-DOC 0.74, p= <0.001) suggesting that water soluble iodine and Br at least partly existed as dissolved organo-halogen compounds. We will add a scatter plot with this correlation. We don't understand the reviewer's comment here. Which correlations are missing, especially regarding the sequential extraction?

10. " Page 9 Line 5: more references are needed. Suggested readings: (Schlegel et al., 2006 Geochimica et osmochimica Acta; Shetaya et al., 2012 Geochimica et Cosmochimica Acta, Li et al., 2017 Journal of Contaminant Hydrology, Xue et al., 2019 Science of the Total Environment, Qian et al., 2020 Science of the Total Environment). "

We will include the references.

11. "Page 10 Lines 1-3 and Lines 6-8: "The solid phase SE analyses showed that between 48 and 152 % (median: 78.7 %) of total iodine (Figure 4) but only between 12.3 % and 72.1 % (median: 30.8 %) of total Br in the soils was associated with reducible components (F4; iron sesquioxides ($Fe_2O_3$), hydroxides ($Fe(OH)_3$) and oxide hydroxides ($FeO(OH)$))" and "This was attributed to higher Fe contents in subsoil horizons, presumably in the form of Fe-oxides" again, the solid phase extraction and speciation of Fe oxide would be needed. "

The exact identification of the Fe-oxides was not aim of our study and would not change the fact that most of the iodine is associated with Fe-Oxides.

12. "Page 11 Fig. 4: the sum of F2, F3, and F4 fractions reaches up to 150%. Please

explain why."

See answer above.

13. "The section about conclusions could be improved; it is rather a summary of results than a real presentation of conclusions. What are the new insights into Fe oxides mediated sorption and mobilization of iodine gained by applying the present fractionation method?"

We will change the conclusion. As stated above, we try to show that iodine is transported as DOC-complexes from topsoil to deeper soil horizons (page 13 Line: 5), where the DOC-iodine (DOC-I) complexes are retained through binding to Fe-oxide surfaces (page 13 Lines: 12-13). The high Fe concentrations in our soils cause the low amounts of leachable iodine (and DOC). Due to the high age of our soils and the long exposure time to iodine depositions the soils had a long time to accumulate iodine in the soil during soil formation due to the process of DOC-iodine leaching from topsoil to subsoil and fixation by Fe-oxides. (page 14 Lines: 11-12). The strong stabilisation by the Fe-Oxides prevent leaching from the soils leading to low iodine concentrations in adjacent drainage systems during base and stormflow conditions.

References

Hou, X., Hansen, V., Aldahan, A., Possnert, G., Lind, O. C., and Lujaniene, G.: A review on speciation of iodine-129 in the environmental and biological samples, Anal Chim Acta, 632, 181–196, https://doi.org/10.1016/j.aca.2008.11.013, 2009.

Kaplan, D. I.: Influence of surface charge of an Fe-oxide and an organic matter dominated soil on iodide and pertechnetate sorption, Radiochim Acta, 91, 75, https://doi.org/10.1524/ract.91.3.173.19977, 2003.

Li, J., Wang, Y., Xie, X., Zhang, L., and Guo, W.: Hydrogeochemistry of high iodine groundwater: A case study at the Datong Basin, northern China, Environ. Sci.: Process. Impacts, 15, 848–859, https://doi.org/10.1039/C3EM30841C, 2013.

[Figure]

Roulier, M., Coppin, F., Bueno, M., Nicolas, M., Thiry, Y., Della Vedova, C., Février, L., Pannier, F., and Le Hécho, I.: Iodine budget in forest soils: Influence of environmental conditions and soil physicochemical properties, Chemosphere, 224, 20–28, https://doi.org/10.1016/j.chemosphere.2019.02.060, 2019.

Schwehr, K. A., Santschi, P. H., Kaplan, D. I., Yeager, C. M., and Brinkmeyer, R.: Organo-Iodine Formation in Soils and Aquifer Sediments at Ambient Concentrations, Environ. Sci. Technol., 43, 7258–7264, https://doi.org/10.1021/es900795k, 2009.

Sheppard, M. I., Thibault, D. H., McMurry, J., and Smith, P. A.: Factors affecting the soil sorption of iodine, Water Air Soil Poll, 83, 51–67, https://doi.org/10.1007/BF00482593, 1995.

Shimamoto, Y. S., Takahashi, Y., and Terada, Y.: Formation of organic iodine supplied as iodide in a soil-water system in Chiba, Japan, Environ. Sci. Technol., 45, 2086–2092, https://doi.org/10.1021/es1032162, 2011.

Unno, Y., Tsukada, H., Takeda, A., Takaku, Y., and Hisamatsu, S.'i.: Soil-soil solution distribution coefficient of soil organic matter is a key factor for that of radioiodide in surface and subsurface soils, J Environ Radioactiv, 169-170, 131–136, https://doi.org/10.1016/j.jenvrad.2017.01.016, 2017.

Xu, C., Miller, E. J., Zhang, S., Li, H.-P., Ho, Y.-F., Schwehr, K. A., Kaplan, D. I., Otosaka, S., Roberts, K. A., Brinkmeyer, R., Yeager, C. M., and Santschi, P. H.: Sequestration and remobilization of radioiodine (129I) by soil organic matter and possible consequences of the remedial action at Savannah River Site, Environ. Sci. Technol., 45, 9975–9983, https://doi.org/10.1021/es201343d, 2011a.

Xu, C., Zhang, S., Ho, Y.-F., Miller, E. J., Roberts, K. A., Li, H.-P., Schwehr, K. A., Otosaka, S., Kaplan, D. I., Brinkmeyer, R., Yeager, C. M., and Santschi, P. H.: Is soil natural organic matter a sink or source for mobile radioiodine (129I) at the Savannah River Site?, Geochim. Cosmochim. Ac, 75, 5716–5735,

https://doi.org/10.1016/j.gca.2011.07.011, 2011b.

Yeager, C. M., Amachi, S., Grandbois, R., Kaplan, D. I., Xu, C., Schwehr, K. A., and Santschi, P. H.: Microbial Transformation of Iodine: From Radioisotopes to Iodine Deficiency, Adv Appl Microbiol, 101, 83–136, https://doi.org/10.1016/bs.aambs.2017.07.002, 2017.

Yoshida, S., Muramatsu, Y., and Uchida, S.: Studies on the sorption of I- (iodide) and IO3- (iodate) onto Andosols, Water Air Soil Poll, 63, 321–329, https://doi.org/10.1007/BF00475499, 1992.

---

## Author Comment (AC3) · 23 Jun 2020

We would like to thank reviewer 3 for the careful assessment of the manuscript and helpful suggestions to improve the quality of our work. Detailed responses to the comments of R3 are given below. The original comments by R3 are between quotation marks.

1. "This paper examines soil pools and hydrologic fluxes of iodine in a tropical forest, in conjunction with other measurements of soil and water chemistry. I concur with the authors' overall interpretation of the data, and I think that the data presented here are potentially useful for the biogeosciences / soil science community, as iodine remains an understudied element. However, I have some significant concerns about details of

data interpretation, specifically in the context of the soil extractions, but believe that these can be remedied in a revision. Specifically, the extractions used here cannot discern whether iodine was in fact directly associated with organic matter vs iron."

We agree with this comment. On page 13 Lines 14-16 we suggested that due to the high affinity of iodine binding to OM most of the iodine in the soil is likely bound to OM and the dissolution of Fe-oxides during step F4 released DOM-bound iodine to Fe-oxides similar as found by Li et al. (2013). Thus, the fraction of iodine bound to OM in deeper soil horizons was likely underestimated. (Page 13 Lines 14-16).

2. "The hydroxylamine extraction only reduces a fraction of the short-range-ordered Fe phases, a fact that is well established in the literature. To extract crystalline Fe, which is likely to be abundant in these soils as demonstrated by the high total Fe content of the subsoil, at least one (if not multiple) extractions with dithionite would be needed. Therefore, attribution of "residual" iodine following extraction by hydroxylamine to an organic-bound (as opposed to Fe-bound) iodine pool is erroneous. "

We agree with this statement and will change this in the revised manuscript. The F5 step will be termed 'residual' including now OM and crystalline Fe-oxides. However, this does not change our main conclusion, that most of the iodine (including DOM-Iodine) was extracted during F4 with hydroxylamine and is associated to SRO Fe phases.

3. "Second, even the iodine released by hydroxylamine may have been proximately bound by organic matter (as suggested by the authors themselves in the discussion). This needs to be made clear earlier in the paper."

We will move this up in the revised manuscript for clarity.

4. "Overall, interpretation of what these extractions mean needs to be more precise and cautious. However, the fact that most of the iodine was typically released in the hydroxylamine extraction is interesting and important and suggests that the iodine was associated with SRO Fe phases and/or organic matter bound with these phases."

We will include a more in-depth discussion of artifacts and misinterpretation of sequential extraction procedures in the revised manuscript. In principal, all sequential extraction procedures include (large) uncertainties arising from re-adsorption to the residue of the extraction step, cross-contamination, incomplete digestion, release of other iodine forms, volatilization or transformation of I, especially in a strong acid/base solution (Shimamoto et al., 2011; Hou et al., 2009). We have already mentioned that in the Manuscript 'In four out of twelve samples the determined sum of the iodine content extracted during F1-F4 were higher (1.4 %,5.8 %,47.7 % and 69.7 %) than the values of the total iodine content caused by cumulated uncertainties of consecutive extractions and inhomogeneity of the soil sample.' (Page 12 Lines 15-17) This probably means that Iodine dissolved during the first, second and third extractions may be re-adsorbed on the active phases/residues of F3 leading to an overestimating of iodine bound to iron oxides. However, from our data we can conclude that only a small amount of iodine is water leachable due to our low iodine concentration in the river water during base and stormflow conditions. Iodine extracted during F4 may also be reabsorbed on the remaining phase (organic matter), which would lead to an underestimation of F4. This would support our findings that most of the iodine is associated to Fe-oxides. As mentioned in the manuscript it is likely that due to the high affinity of iodine binding to OM most of the iodine in the soil is bound to OM and the dissolution of Fe-oxides during step F4 released DOM-bound iodine to Fe-oxides similar as found by Li et al. (2013). Thus, the fraction of iodine bound to OM in deeper soil horizons was likely underestimated. (Page 13 Lines 14-16). Despite all the uncertainties, most of the iodine was extracted during the hydroxylamine extraction suggesting that most of the iodine in our soils is associated to SRO Fe phases and/ or OM bound to it as also stated by the third reviewer. We also believe that the exact separation between Fe-oxide bound and organically bound iodine is not the essential point as these components never exist completely separated in soils. We try to show that the high retention of iodine through adsorption of DOM-iodine complexes to Fe-oxides is the major process of iodine enrichment in tropical soils and the resulting low iodine concentrations in adjacent

drainage systems. This is to our knowledge the novelty in this study. We will make this point clearer in the revised manuscript.

5. "If the DOC content of the hydroxylamine extraction was measured, the authors could assess whether DOC was adsorbed or coprecipitated using the molar DOC/Fe ratios."

Unfortunately, we have not measured this to prevent instrument damage.

6. "Second, the rationale for sampling the different soil profiles to different depths was not at all clear. Given that bedrock was not present, why were some profiles only sampled to very shallow depths? This complicates our interpretation of the data. It is difficult to interpret a depth profile of only two measurements in terms of transport/retention dynamics of an element. Some of the "A" horizons are clearly "O" horizons based on high concentrations of C."

The study was conducted in a Biological Reserve in Costa Rica, which underlies stricter sample extraction regulations than national parks. Additionally, very steep slopes and heavy root penetration complicated the soil sampling. Therefore, it was unfortunately not possible to collect soil samples from greater depths. We agree that the depth profiles with only two data points are difficult to interpret and can only serve as an orientation. However, other hydrogeomorphological studies (e.g., Dehaspe et al., 2018) support the notion that only the upper soil horizons above 1m depth play a role in runoff generation that can be detected in the stream network based on biogeochemical tracers. Concerning the topsoil horizons with high C concentrations, we assume you are referring to profile L2 with 33.8 % carbon. Despite >30 % carbon, we decided to classify this horizon as an A horizon due to its low thickness (12 cm).

7. "Overall, the writing was not well-focused and often rambled. For example, the first two paragraphs of page 2 need clear topic sentences and organization to guide the reader through an argument. "

We will improve focus in our writing as suggested.

8. "The rationale for studying I and Br together should be introduced, since these trends become a major part of the results. "

Will be changed in the revised manuscript.

9. "In the introduction, it would be helpful to briefly describe the biogeochemistry of iodine in a bit more detail, because this is an uncommon topic in the literature. Some of these topics are addressed in the discussion. (e.g. differences in binding of iodine and bromine). For example, what are the major species of iodine in soil? What kinds of bonds do they form with SOM and the major mineral phases?" We will include a more detailed description in the introduction of the revised manuscript.

10. "Specific comments: L17: "Stream water was sampled randomly over a period of five weeks": this is haphazard, not random sampling I believe?"

The sampling took place every second day or even daily in four intervals each of three until five days covering base and storm flow conditions. The time of sampling was depending on the accessibility to the research station located in a pristine rainforest.

11. "P1 Introduction: discussion of the health impacts of iodine seems remotely linked to the focus of this paper; better to make this explicit or remove" We decided to remove this from the revised manuscript. 12. P2 L22: Avoid overgeneralizing about "tropical soils". Note that almost all of the global soil orders (except Gellisols) can be found in the tropics. Clarify the scope of your study accordingly. Thank you for your comment on this, we agree that the term "tropical soils" should be specified. 13. "P3: "(REF to the classification system)" – revise or delete Given your previous description, it seems as if this soil is really a Ferralsol?" The classification of the soils is difficult due to the shallow sampling depths and limited analysis of the mineral phase. However, from field observations there was no dominance of hematite (also cf. Munsell soil colour), which is a common iron mineral in ferralsols. The soils were classified using the World

Reference Base for Soil Resources (IUSS Working Group WRB, 2015). We will include the reference.

14. "P4: Why did your sampling depth vary among soil pits (e.g. 0.5 m or 1 m)" Due to strict regulations in the biological reserve, steep slopes and heavy rooting, we decided to only sample two soil pits to 1 m and sample all other soil pits to 0.5 m.

15. "P5: The water sampling scheme is unclear. Did your sampling span both base flow and stormflow conditions?"

The streamwater sampling took place every second day or even daily in four intervals each of three till five days covering base and storm flow conditions. The time of sampling was depending on the accessibility to the research station in a pristine rainforest.

16. "P6: Note that only a fraction of soil Fe phases are reduced with hydroxylamine. Dithionite is needed to reduce crystalline Fe. Furthermore, the hydroxylamine extraction will also extract iodine bound with organic matter, because Fe-associated organic matter is released. Therefore, the F5 fraction cannot be used to represent OM-bound iodine, as substantial iodine may remain associated with Fe (and other mineral) phases, and previous extractants (e.g. F4) likely included OMbound iodine. See for example Coward et al. 2017 10.1016/j.geoderma.2017.07.026"

See above

17. Section 2.3.2: what kind of water was used for the leaching experiments?

MilliQ Water (18.2 M$\Omega$.cm).

18. "P8 L8: What kind of "temperate soils", and how representative are these? Note the tremendous diversity in soil types and likely iodine input/output budgets among ecoystems."

Above all, we wanted to point out the climatic and age effects between soils in tropical and temperate regions. High age and continuously high rainfall resulted in long term intense chemical weathering of bedrock and soils, which lead to highly weathered acidic soils with high accumulation of Fe-Oxides. The weathering of primary minerals is the same process in temperate and tropical climates, differing only in its greater intensity in the tropics.

Here a short list of some studies. • (Korobova, 2010): Russian plain forest; • (Muramatsu et al., 2004): Chiba Prefecture, Pacific side of Japan (Humid subtropical climate); • (Roulier et al., 2018; Roulier et al., 2019): French beech forest soils; • (Takeda et al., 2015; Takeda et al., 2018): pine forest, Japanese beech forest and dwarf bamboo lowland (Rokkasho, Japan (Southern Shimokita Peninsula; cold maritime climate)

19. "P9 L5: What "previous studies"? Present your data first. You need to cite specific literature if you want to compare."

We will change the order and present our data first

20. "P9 L7: This organic C concentration is too high for a mineral A horizon. This indicates that an O (organic) horizon was sampled."

As mentioned above, we classified the topsoil horizon in profile L2 as an A horizon instead of an O horizon due to the low thickness of the horizon (12 cm).

21. "P10: Note that the Fe-associated iodine is underestimated because your extractions did not release crystalline Fe phases, which likely dominated here (especially in the subsoil) Figure 4: It is concerning that three samples do not have any F5 fraction (difference between total and extracted iodine). In how many cases was this value negative (e.g. more iodine was extracted than in the total measured sample)"

'In four out of twelve samples the determined sum of the iodine content extracted during F1-F4 were higher (1.4 %,5.8 %,47.7 % and 69.7 %) than the values of the total iodine content caused by cumulated uncertainties of consecutive extractions and inhomogeneity of the soil sample'. Page 12 Lines 15-17 We agree that crystalline Fe

oxides were not extracted during F4, iodine bound to crystalline Fe oxides is part of the residual fraction F5.

22. "P12 L1: Following the reasoning above, you cannot determine that F5 fraction iodine is associated with SOM." See above, we defined the F5 fraction now as residual-fraction including now both SOM and crystalline Fe.

23. "P14 L1-5: The occurrence of Fe reduction and DOC/nutrient mobilization in tropical soils has received significant recent attention, even (and especially) in systems with high rainfall and high infiltration rates. Iron reduction is widespread in these kinds of ecosystems. It would help to read and cite relevant literature here. P15 L6 "Moreover, the low mobility of iodine as DOC-I-Fe-oxide- complex was caused by the fact that Fe-oxides protect OM against degradation." Note recent findings that challenge this notion; Fe/C/nutrient interactions can be dynamic."

Dehaspe et al. (2018) and Solano-Rivera et al. (2019) have shown that our catchment is rapidly responding to rain events. They found saturation excess overland flow during periods with high rainfall since soils throughout the catchment have high infiltration capacities in excess of 200 mm/h (Solano-Rivera et al., 2019) or even up to >1000 mm/h in places (Dehaspe et al., 2018). It is possible that anoxic conditions are temporarily reached (Chen et al., 2020). However, the high iodine content in our soils shows that oxic conditions or reducing conditions of nitrate and manganese reduction predominate anoxic conditions of iron reduction. Another response may be found in flatter slowly responding systems.

References

Chen, C., Hall, S. J., Coward, E., and Thompson, A.: Iron-mediated organic matter decomposition in humid soils can counteract protection, Nature communications, 11, 2255, https://doi.org/10.1038/s41467-020-16071-5, 2020.

Dehaspe, J., Birkel, C., Tetzlaff, D., Sánchez-Murillo, R., Durán-Quesada, A. M., and

[Figure]

Soulsby, C.: Spatially distributed tracer-aided modelling to explore water and isotope transport, storage and mixing in a pristine, humid tropical catchment, Hydrol. Process., 110, 5089, https://doi.org/10.1002/hyp.13258, 2018.

Hou, X., Hansen, V., Aldahan, A., Possnert, G., Lind, O. C., and Lujaniene, G.: A review on speciation of iodine-129 in the environmental and biological samples, Anal Chim Acta, 632, 181–196, https://doi.org/10.1016/j.aca.2008.11.013, 2009.

IUSS Working Group WRB: World reference base for soil resources 2014, update 2015: International soil classification system for naming soils and creating legends for soil maps, World soil resources reports, 106, FAO, Rome, 181 pp., 2015.

Korobova, E.: Soil and landscape geochemical factors which contribute to iodine spatial distribution in the main environmental components and food chain in the central Russian plain, J Geochem Explor, 107, 180–192, https://doi.org/10.1016/j.gexplo.2010.03.003, 2010.

Li, J., Wang, Y., Xie, X., Zhang, L., and Guo, W.: Hydrogeochemistry of high iodine groundwater: A case study at the Datong Basin, northern China, Environ. Sci.: Process. Impacts, 15, 848–859, https://doi.org/10.1039/C3EM30841C, 2013.

Muramatsu, Y., Yoshida, S., Fehn, U., Amachi, S., and Ohmomo, Y.: Studies with natural and anthropogenic iodine isotopes: Iodine distribution and cycling in the global environment, J Environ Radioactiv, 74, 221–232, https://doi.org/10.1016/j.jenvrad.2004.01.011, 2004.

Roulier, M., Coppin, F., Bueno, M., Nicolas, M., Thiry, Y., Della Vedova, C., Février, L., Pannier, F., and Le Hécho, I.: Iodine budget in forest soils: Influence of environmental conditions and soil physicochemical properties, Chemosphere, 224, 20–28, https://doi.org/10.1016/j.chemosphere.2019.02.060, 2019.

Roulier, M., Bueno, M., Thiry, Y., Coppin, F., Redon, P.-O., Le Hécho, I., and Pannier, F.: Iodine distribution and cycling in a beech (Fagus sylvatica) temperate forest, Sci.

Total Environ., 645, 431–440, https://doi.org/10.1016/j.scitotenv.2018.07.039, 2018.

Shimamoto, Y. S., Takahashi, Y., and Terada, Y.: Formation of organic iodine supplied as iodide in a soil-water system in Chiba, Japan, Environ. Sci. Technol., 45, 2086–2092, https://doi.org/10.1021/es1032162, 2011.

Solano-Rivera, V., Geris, J., Granados-Bolaños, S., Brenes-Cambronero, L., Artavia-Rodríguez, G., Sánchez-Murillo, R., and Birkel, C.: Exploring extreme rainfall impacts on flow and turbidity dynamics in a steep, pristine and tropical volcanic catchment, CATENA, 182, 104118, https://doi.org/10.1016/j.catena.2019.104118, 2019.

Takeda, A., Tsukada, H., Takahashi, M., Takaku, Y., and Hisamatsu, S.: Changes in the chemical form of exogenous iodine in forest soils and their extracts, Radiation protection dosimetry, 167, 181–186, https://doi.org/10.1093/rpd/ncv240, 2015.

Takeda, A., Nakao, A., Yamasaki, S.-i., and Tsuchiya, N.: Distribution and Speciation of Bromine and Iodine in Volcanic Ash Soil Profiles, Soil Sci. Soc. Am. J., 82, 815–825, https://doi.org/10.2136/sssaj2018.01.0019, 2018.